# Utilizing mentorship education to promote a culturally responsive research training environment in the biomedical sciences

**Sarah Suiter**[1]*, **Angela Byars-Winston**[2], **Fátima Sancheznieto**[3], **Christine Pfund**[3], **Linda Sealy**[4]

**1** Department of Human & Organizational Development, Vanderbilt University, Nashville, Tennessee, United States of America, **2** Department of Medicine, Institute for Diversity Science, University of Wisconsin-Madison, Madison, Wisconsin, United States of America, **3** Center of the Improvement of Mentored Experiences in Research (CIMER), Wisconsin Center for Education Research, University of Wisconsin-Madison, Madison, Wisconsin, United States of America, **4** School of Medicine Basic Sciences, Vanderbilt University, Nashville, Tennessee, United States of America

* sarah.v.suiter@vanderbilt.edu

**Data Availability Statement:** De-identified quantitative data from participant surveys has been made accessible via Zenodo at https://zenodo.org/

## Abstract

There is an urgent and compelling need for systemic change to achieve diversity and inclusion goals in the biomedical sciences. Because faculty hold great influence in shaping research training environments, faculty development is a key aspect in building institutional capacity to create climates in which persons excluded because of their ethnicity or race (PEERs) can succeed. We present a mixed methods case study of one institution's efforts to improve mentorship of PEER doctoral students through mentorship education workshops for faculty. These workshops were one strategy among others intended to improve graduate trainees' experiences, and positively affect institutional climate with respect to racial and ethnic diversity. Surveys of 108 faculty mentors revealed that about 80% overall agreed or strongly agreed with the value of culturally responsive mentoring behaviors but about 63% overall agreed or strongly agreed that they were confident in their ability to enact those behaviors. Through a series of three focus groups, PEER doctoral students reported that they noticed mentors' efforts to address cultural diversity matters and identified some guidance for how to approach such topics. We discuss future directions and implications for using mentorship education to activate systemic change toward inclusive research training environments and promoting the value of mentorship within institutions.

## Introduction

Lack of racial and ethnic diversity in science, technology, engineering, and math (STEM) fields is a well-documented problem [1–3]; one that begins in undergraduate education and worsens as students advance through their academic pathways and into STEM careers [4]. The National Center for Science and Engineering Statistics found that persons excluded from science and higher education because of their ethnicity or race (PEERs) enter college with similar or higher

doi/10.5281/zenodo.10372674. Qualitative data from pre-participation surveys and from focus group transcripts will not be made publicly available in order to protect participants from potential identification.

**Funding:** SS: [NIH-5U54CA163072] National Institutes of Health, National Cancer Institute https://www.cancer.gov LS: [NIH-5U54CA163072] National Institutes of Health, National Cancer Institute https://www.cancer.gov [NIH-P30CA068485] National Institutes of Health, National Cancer Institute https://www.cancer.gov [NIH-5R25GM062549] National Institutes of Health https://www.nih.gov The funders had no role in study design, data collection and analysis, decision to publish, or preparation of the manuscript.

**Competing interests:** The authors have declared that no competing interests exist

rates of intention to pursue a STEM degree, but their representation diminishes at each stage of higher education [2, 5, 6]. For example, in 2019, PEERs constituted 30% of the US population, but earned 18% of STEM bachelor's degrees and 11% of STEM PhDs [2]. Underrepresentation persists in the US scientific workforce, where 13% of the STEM workforce and only 4% of faculty at research institutions identify with a minoritized racial or ethnic group [3]. These disparities are consequential for STEM fields themselves, as racial diversity is associated with increased innovation and other forms of scientific excellence [7]. They are also evidence of long-standing injustices that higher education has an obligation to address [5, 8, 9].

Investing in robust, culturally responsive mentoring education and training for faculty has been used as a strategy to improve student experiences and drive change with respect to institutional policies, practices, and culture [10]. The National Academies of Science, Engineering, and Medicine (NASEM) report, *The Science of Effective Mentorship in STEMM*, defines mentorship as a "professional, working alliance in which individuals work together over time to support the personal and professional growth, development, and success of the relational partners through the provision of career and psychosocial support" (p.37) [8]. As highlighted in the report, mentoring relationships play a pivotal role in student academic outcomes such as degree completion, academic productivity, and job acquisition, as well as psychosocial outcomes like sense of belonging, anxiety, and depression [8, 11]. Effective mentors help students access resources and navigate institutional systems [6], create positive micro-climates within their own laboratories and research teams [12, 13], and play a direct role in socializing students to STEM fields and careers [14]. The impact of mentorship is even more pronounced for PEERs studying at predominantly white institutions [15]. Most faculty, however, do not receive formal mentorship education, and many institutions do not provide the types of requirements and/or incentives that communicate to faculty and students that mentorship is something they value [11]. Nevertheless, mentorship education has the potential to improve effective mentorship knowledge and practices, promote the value of mentorship within institutions, and activate systemic change toward an inclusive science [16–18].

We conducted a study of one institution's efforts to improve mentorship of PEER doctoral students as a strategy to improve graduate trainees' experiences, and as a strategy to positively affect institutional climate with respect to racial and ethnic diversity. Specifically, we used a mixed methods case study [19], that included surveys administered to faculty who participated in a mentorship education intervention designed to teach and reinforce culturally responsive mentoring practices and a series of three focus groups involving PEER doctoral students whose mentors had participated in the intervention. Our case study was guided by the following three research questions: 1) What types of mentoring experiences have faculty had? 2) How did faculty respond to the mentorship education? 3) How did students perceive faculty mentoring?

Furthermore, we situated our data analysis and interpretation in Hurtado and colleagues' organizational learning model [20] to illustrate how systemic change at the graduate department level in STEM disciplines may be catalyzed by faculty who participate in institutional-level faculty development interventions. Their model begins with science faculty gaining new knowledge about the experiences of trainees, especially those from PEER groups, followed by interventions that can change faculty mindsets and behaviors. In turn, diffusion of new knowledge can lead to critical questioning of existing departmental and campus policies and practices that can be hostile to PEER students. Ultimately, critical questioning of the institutional status quo research training practices necessitates greater faculty buy-in to overcome resistance due to deeply embedded elements of culture and power dynamics that must be confronted to sustain new inclusive practices aimed at student success.

Our case study is distinct from previous studies on mentoring interventions in STEM in at least two ways. First, unlike many mentor education programs in which participants are selected for participation based on their experience with, commitment to, and talent for mentoring prior to the training, this training was open to all faculty in specific departments and programs at various time points. As a result, we are able to explore variations in motivation for faculty participation, experience with mentoring PEER students, and perceptions of mentorship in a more generalized group of faculty located in one institution. Second, the use of qualitative data collection strategies to gain the perspectives of mentees themselves provides rich information with respect to student experiences of mentorship as their mentors implement improvements to their mentoring practices. We were able to gain insight into students' expectations for mentoring, qualities of positive and negative mentoring experiences, and student perspectives on institutional drivers of mentoring quality.

## Methods

### Case study context

We conducted our case study with The Basic Sciences of the Vanderbilt School of Medicine and the research enterprise of Vanderbilt University Medical Center. Together, they have a 20 + year commitment to diversity in graduate education. Overall, Vanderbilt has awarded 263 doctoral degrees in biomedical research to students from PEER groups since 1997, which represents 13% of the total PhDs awarded as of May 2024. The NIH-funded Initiative for Maximizing Student Diversity (IMSD) program at Vanderbilt, which began in 2007, has continuously increased the proportion of PEER students completing doctoral degrees. For example, in the past five years (2020–2024) 390 PhDs have been awarded with 21% (or 83 PhDs) completed by PEER students.

Nonetheless, results from a Graduate Student Mentoring, Climate, and Career Plans Survey of Vanderbilt doctoral students across 13 different graduate programs in the biomedical sciences conducted in 2017 revealed concerns. Survey results highlighted disparities between PEER and non-PEER doctoral students in areas such as feelings of isolation in their department or program, concerns over work life balance, and lower assessment of opportunities to discuss how their racial/ethnic or gender influences their training experience. At Vanderbilt, like many research-intensive US universities, a PhD student's primary advisor/mentor plays a sizable role in the student's doctoral education. The mentor typically leads the lab in which the student works, directs, supports and evaluates the student's research, and serves as a member of the student's dissertation committee. Mentors, then, have the ability to affect much about a doctoral student's experience. It was within this context that the plan for mentorship education for faculty within the Vanderbilt Basic Sciences emerged.

**Mentorship education workshop delivered to vanderbilt basic sciences faculty.**    For our mentorship education workshop, we committed to the NASEM concept that effective mentorship is necessarily culturally responsive whereby mentors "show curiosity and concern for students' cultural backgrounds" (pg. 62) and their non-STEM social identities [8]. We followed a mentorship education model similar to that implemented by three of the authors of this paper for the Howard Hughes Medical Institute (HHMI) Gilliam Fellowship mentors [16]. The curriculum facilitates mentorship skill development across two full days. The curriculum includes: 1) foundational mentorship education that covers specific competencies and characteristics of effective instrumental and psychosocial mentorship roles; 2) extended focus on building mentors' cultural responsiveness to address diversity factors in their research mentoring relationships; and 3) targeted work on considering mentoring relationship dynamics within the local research training environments. Based on the lack of racial and ethnic diversity in STEM fields

noted in the Introduction section of this paper and in other publications [1–3, 15], the Vanderbilt mentorship workshop focused largely on racial and ethnic diversity matters in research mentoring relationships.

The Vanderbilt mentorship education curricular elements are informed by well-studied, culturally informed mentor training interventions from the following curricula: *Entering Mentoring* [21–24], *Optimizing the Practice of Mentoring* [25], *Promoting Mentee Research Self-Efficacy* [26], and the *Culturally Aware Mentoring (CAM) Workshop* [27, 28]. More detailed information about the Entering Mentoring and Culturally Aware Mentoring curricula can be found at http://www.CIMERproject.org. Table 1 provides an overview of the number of people who participated in the Vanderbilt mentorship education workshops, workshop dates, and the formats of the workshop for six cohorts from 2017–2021.

We sought and were granted study approval by the University of Wisconsin and Vanderbilt University Institutional Review Boards. Written informed consent was obtained from participants for the surveys and focus groups; informed consent documents were administered electronically.

## Data collection and participants

Data were collected through two primary methods: surveys with faculty participants and focus groups with graduate students whose mentors had participated in mentorship education workshops. S1 File includes our study measures (survey and focus group guides) and information regarding how to access deidentified survey data.

**Surveys with faculty workshop participants.** We collected data from participating faculty mentors before the workshop (for planning and preparation purposes) and immediately following the workshop. Data were collected using Qualtrics survey platform. Questions included assessment of program components, including participant satisfaction, learning, mentorship quality, and outcomes, including the Cultural Diversity Awareness (CDA) measure with three subscales assessing attitudes, behaviors, and confidence to enact behaviors reflecting CDA in mentoring relationships [28]. Enrollment periods for mentor education planning surveys began two weeks before the two-day workshops. Post-participation surveys were open for approximately two weeks after the workshops, with the exception of the first year, in which the post-surveys were sent about one month after the workshop. Workshop training dates were November 28–29, 2017; April 9–10, 2018; December 4–5, 2018; December 5–6, 2019;

**Table 1. Participants in the vanderbilt mentorship education workshops 2017–2021.**

| Description of Participants | # of Participants | Date of Training | Format of Training | Contact Hours |
|---|---|---|---|---|
| Cohort 1: By invitation - Leadership of School of Medicine (SOM) Basic Sciences (Dean, Assoc. Deans, Dept Chairs) and faculty with key roles in biomedical PhD education | 27 | 11/2017 | In-person | 16 |
| Cohort 2: By invitation - Leaders in Vanderbilt Cancer Center, faculty involved in Meharry-Vanderbilt-TSU Cancer Partnership, faculty in Vanderbilt Cancer Biology graduate program. | 21 | 4/2018 | In-person | 16 |
| Cohort 3: Open call to all SOM faculty involved in training PhD graduate students, past, present, or future. | 25 | 12/2018 | In-person | 16 |
| Cohort 4: Open call to all SOM faculty involved in training PhD graduate students, past, present, or future. | 32 | 12/2019 | In-person | 16 |
| Cohort 5: Open call to SOM Basic Sciences faculty who had not yet attended. Strongly encouraged (but not required) by their Dept Chair to attend. | 31 | 12/2020 | On-line, Synchronous | 8 |
| Cohort 6: Open call to all SOM faculty involved in training PhD graduate students, past, present, or future. | 32 | 12/2021 | In-person | 12 |

December 1–4, 2020; and December 6–7, 2021. All faculty who participated in the mentorship education workshops were invited to take the surveys.

**Focus groups with doctoral students.** Co-author S.S. conducted three successive focus groups with students participating in the IMSD program who had been in their labs before and after their mentor participated in the mentorship workshop. First and second year students were excluded from the study because of the impact of the COVID-19 pandemic on their early graduate school experiences. Specifically, we believed their experiences with mentoring, course work, and lab work were meaningfully different enough from previous students to justify their exclusion from this study. Each focus group involved the same set of students, and students responded to different questions in each group. S.S. had no relationship with the students prior to their participation in the study. The focus groups were held on-line, via Zoom, and were spaced approximately three days apart. The first group focused on participants' expectations regarding mentoring when they entered graduate school. In the second focus group, we asked students to reflect on their current mentoring experiences, and whether those experiences met their initial expectations. In the third focus group, we asked students to recommend changes that they thought would improve mentoring of students at Vanderbilt, including but not limited to PEER students. Each focus group lasted approximately one hour. Focus group enrollment period began on May 17, 2021, when all eligible students received an email inviting them to participate in the focus groups. Participants were asked to respond by May 26, 2021, if they were interested. All interested students were invited to participate and confirmed their participation on the day of the first focus group during the informed consent process. Focus groups were held June 1, 3, and 8, 2021. Otter.ai was used to generate transcripts of the focus groups while they were conducted on zoom. Focus groups were not video recorded. Immediately after each focus group, S.S. checked the transcripts against her notes for accuracy and de-identified the transcripts.

## Data analysis

We focused our analyses of the post-participation survey data on faculty participants' values related to mentoring, mentoring confidence, use of specific mentoring behaviors, and satisfaction with the mentoring education they received. We also include an analysis on one question from the pre-participation survey which asked: Why did you choose to sign up for this workshop? These data were analyzed for all six faculty participant cohorts 2017–2021. Our analysis of mentee focus group data explores mentees' perceptions of the mentoring they received at Vanderbilt, while also examining the broader context and IMSD-driven changes mentees experienced.

**Quantitative analysis.** Post training survey data from across cohorts were cleaned and combined using Microsoft Excel. Boxplots were generated using the ggplot2 package (https://ggplot2.tidyverse.org/) in R version 4 and Pearson's bivariate correlations for CDA mean scale scores using SPSS version 28.

**Qualitative analysis.** Qualitative data from open-ended survey questions were analyzed using an iterative coding methodology [29]. For the first round of coding, open coding was employed by co-author C.P. In the second round, co-authors F.S. and C.P. worked collaboratively to consolidate codes and ensure they were applied consistently. Focus group data were analyzed in HyperResearch qualitative analysis software (http://www.researchware.com) by co-author S.S. using a grounded theory approach [30] that involves reading transcripts and assigning codes to words or phrases in the transcripts and then exploring the prevalence and relationships among codes to develop concepts and theories from the data. S.S. then presented her analysis to the rest of the research team in written format, including codes, themes, and

extensive quotations. The group discussed the analyses via email and at future meetings, offering questions and alternative interpretations. Ultimately, the group reached agreement on codes, themes, and their applications to the qualitative data. Thus, the research team served in the role of "critical friends," a strategy known for its ability to enhance the rigor and accuracy of qualitative data analysis [31]. When reporting qualitative data, we used participant identification numbers rather than names or demographic descriptors to maintain the confidentiality of the students. Additionally, we have used the gender-neutral pronoun "they" (versus "he" or "she") when describing students to further conceal their identities. There are times when participant responses reveal their gender. When these revelations are important to understanding the participant quotation, we have left them in. When students are discussing their mentors, we have retained demographic descriptions when those descriptions are important to understanding the data.

## Results

Of the 168 workshop participants, 108 completed the post-surveys and consented to their use for research purposes, yielding a survey response rate of 64% for faculty returning post-surveys. Table 2 provides demographic descriptors of faculty participants by year.

### Faculty experiences with mentoring and mentorship education

Faculty participants (n = 108) had varied experience with mentoring graduate students, ranging from faculty who had never mentored a graduate student (3.7%, n = 4) to faculty who had mentored more than 11 graduate students (35.1%, n = 38). In general, workshop participants had extensive experience with mentoring, with 63% (n = 68) indicating that they had mentored 6 or more graduate students. These trends were reversed when faculty were asked about their experience mentoring graduate students from PEER groups. Fifty-eight percent (n = 63) of faculty reported that they had mentored two or fewer students from PEER groups, with only 7.4% (n = 8) reporting having mentored 11 or more. Despite the vast majority of participants having mentored at least one graduate student, only 44% (n = 48) of faculty reported having previously participated in mentor training of any kind.

### Faculty motivation for participating in mentorship education

Ninety-four people responded to an open-ended prompt on the pre-participation survey that inquired, "Why did you sign up for the workshop?" The responses revealed a number of motivations, such as desire to improve ability to mentor trainees from diverse backgrounds (40%, n = 38), desire to improve mentoring skills generally (27%, n = 25), being invited or strongly encouraged to attend (12%, n = 11), and preparing for a new campus role (4%, n = 4). Although most participants appeared to have taken part in the workshops because they were motivated to improve their mentoring in one way or another, some participated in response to external influences. For example, one participant indicated they attended because, "my university now considers having such training as necessary to allow PIs to mentor PEER students." Another said they attended because of "influence from a local colleague invested in such matters." Table 3 displays themes in participant responses around motivation for participating, as well as counts and sample quotations.

### Faculty perceptions of their mentoring effectiveness

In response to items intended to assess participant perspectives on their own mentoring effectiveness and quality following the mentorship education workshop, faculty rated their

Table 2. Sociodemographic characteristics of workshop participants.

| Sample Characteristics | n | % |
|---|---|---|
| Gender | | |
| Man | 51 | 47 |
| Woman | 46 | 43 |
| Did Not Report | 11 | 10 |
| Race | | |
| White | 79 | 73 |
| Asian | 10 | 9 |
| Black or African American | 7 | 6 |
| American Indian or Alaskan Native | 1 | 1 |
| Native Hawaiian or Pacific Islander | 1 | 1 |
| Other race not listed | 3 | 3 |
| Did not report | 10 | 10 |
| Ethnicity | | |
| Hispanic/Latino | 7 | 6 |
| Not Hispanic or Latino | 84 | 78 |
| Did Not Report | 17 | 16 |
| Title | | |
| Assistant Professor | 32 | 30 |
| Associate Professor | 21 | 19 |
| Professor | 43 | 40 |
| Assistant Dean, Associate Dean, or Dean | 7 | 6 |
| Training Program Director | 6 | 5 |
| Other Title Not Listed | 8 | 7 |
| Did Not Report | 9 | 8 |

Note N = 108; Participants could select more than one race and more than one title

mentoring skills favorably across participant cohorts. Fig 1A demonstrates response distribution (Likert-type scale 1 [strongly disagree] to 4 [strongly agree]) across all six cohorts to questions regarding mentoring effectiveness including the mentor's showing interest in student projects (n = 103), and their efforts to make students feel included in their lab and lab culture (n = 102). Although there are some faculty who responded "strongly disagree (1)" or "disagree (2)" in a few of the participant cohorts, faculty overwhelmingly either strongly agreed or agreed (101, 98% for showing interest; 99, 97% for inclusion) with both items. Similarly, faculty rated themselves highly on items intended to measure perceptions of mentoring quality. Across the participant cohorts, the majority of faculty rated their working relationship with their mentees as good (61, 60%) or excellent (39, 38%) and the overall quality of their relationships as good (67, 65%) or excellent (29, 28%) as well. Fig 1B displays response distribution across all six cohorts.

**Cultural diversity awareness (CDA) in mentoring: Attitudes, confidence, and behaviors.** Like previous implementations of culturally responsive mentorship education [16, 17] faculty reported high levels of agreement with CDA attitudes scale items such as, "*It is important for mentors and mentees to talk together about the mentee's racial/ethnic background*," and "*It is important for mentors and mentees to discuss how race/ethnicity impacts the mentee's research experience*." Figs 2 and 3 display response distribution across all six cohorts for CDA attitudes, confidence, and behaviors subscales. A Kruskal-Wallis test revealed that cohorts were statistically indistinguishable overall from one another when comparing means and

**Table 3. Faculty motivation for participating in the workshop.**

| Themes | Number of respondents | Example quotes |
|---|---|---|
| Increase ability to mentor trainees from diverse background/ increase cultural awareness | 38 | • *To better understand mentoring of individuals with diverse backgrounds, so that I can optimally run a research program that engages diversity towards creativity* <br> • *To improve my awareness/thinking of diversity and culture identity and my mentoring effectiveness.* |
| Improve mentoring skills, generally | 25 | • *I am trying to improve my mentorship. I want to be a better mentor, and I feel I need help to do that.* <br> • *To be better educated on the topic.* <br> • *Learn about new/different approaches for individualized mentoring* |
| Invited or strongly encouragement to attend | 11 | • *I was asked to sign up as a director of graduate studies.* <br> • *Strong recommendations to attend from [Blinded institution] leadership* <br> • *Influence of a local colleague invested in such matters* |
| Required to attend | 7 | • *Required for training grants* <br> • *My university now considers having such training as necessary to allow PIs to mentor URM students.* |
| Outcomes of past mentees | 4 | • *My mentees from historically underrepresented racial backgrounds do not perform as well as those who are white. I assume that this is a weakness of my mentoring.* |
| Gain perspectives of other mentors | 4 | • *To gauge where I stand as a research mentor in general and gain an understanding as to where my peers are to and what we can be doing better.* |
| Prepare for a new campus role | 4 | • *Have recently taken on responsibilities for the . . . which requires my oversight of the career development of nearly thirty trainees.* <br> • *I am now "mentoring mentors" as a division leader with new responsibilities for faculty and facilitating their ability to mentor.* |
| Curiosity | 3 | • *Curiosity. I want to know what else is out there.* |

Note Response Rate = 57% (94/165)

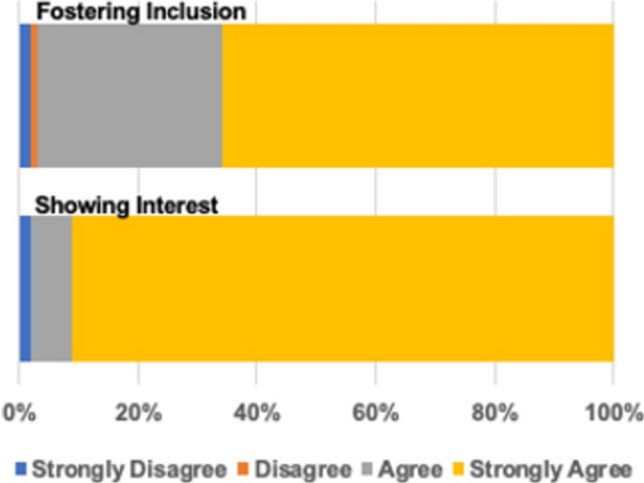
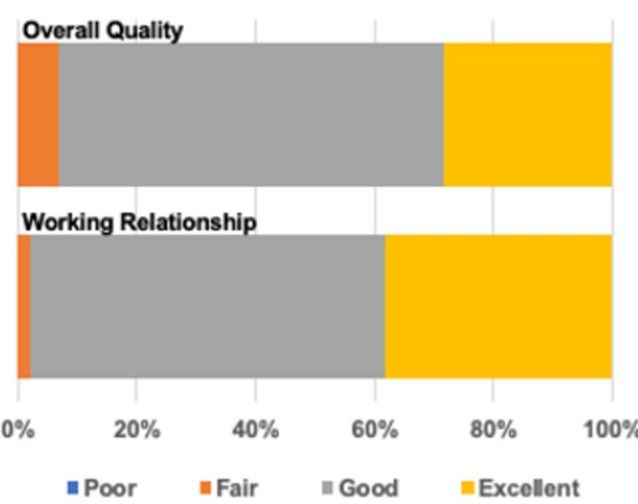

**Fig 1. Self-reported mentoring effectiveness and quality.** A. Self-reported mentoring effectiveness following training for all six cohorts. Mentors rated their agreement using a Likert-type scale from 1 (strongly disagree) to 4 (strongly agree) on the following items: "I made my mentee feel included in the lab (or field setting)" [Fostering Inclusion]; "I tried to show interest in my mentee's projects" [Showing Interest] B. Self-reported mentoring quality following training for all six cohorts. Mentors rated the quality of two aspects of their mentoring relationship using a Likert-type scale ranging from (1) Poor to 4 (Excellent): "The overall quality of my research mentorship relationship" [Overall Quality]; "My working relationship with my research mentees" [Working Relationship].

distributions of individual items and, thus, we have collapsed data across all cohorts for simplicity in viewing and interpreting figures. Two items, however, were statistically different across the cohorts: "My racial/ethnic identity is relevant to my research mentoring relationships," (H(5) = 12.206, $p$ = 0.032) and "I intentionally created opportunities for my mentees to bring up issues of race/ethnicity as they arose" (H(5) = 12.972, $p$ = 0.024). These differences for two specific items may be due to varying reasons. Cohort 2 had slightly lower scores for "racial/ethnic identity is relevant. . ." which may be due to the unique perspectives of the individuals who participated at that time. Cohort 6 had higher scores than the other cohorts on the "intentionally created opportunities" item. This cohort met in 2021 after COVID-19 and the racial justice protests of 2020. Thus, their comparatively higher endorsement of this item may reflect the period effect of increased public discourse on race at that time in the US.

Faculty ratings of their confidence to enact CDA behaviors were comparatively lower in absolute value than ratings of their attitudes toward the relevance of CDA in mentorship. This is true for all questions and all cohorts, except for the item "Notice interactions in the

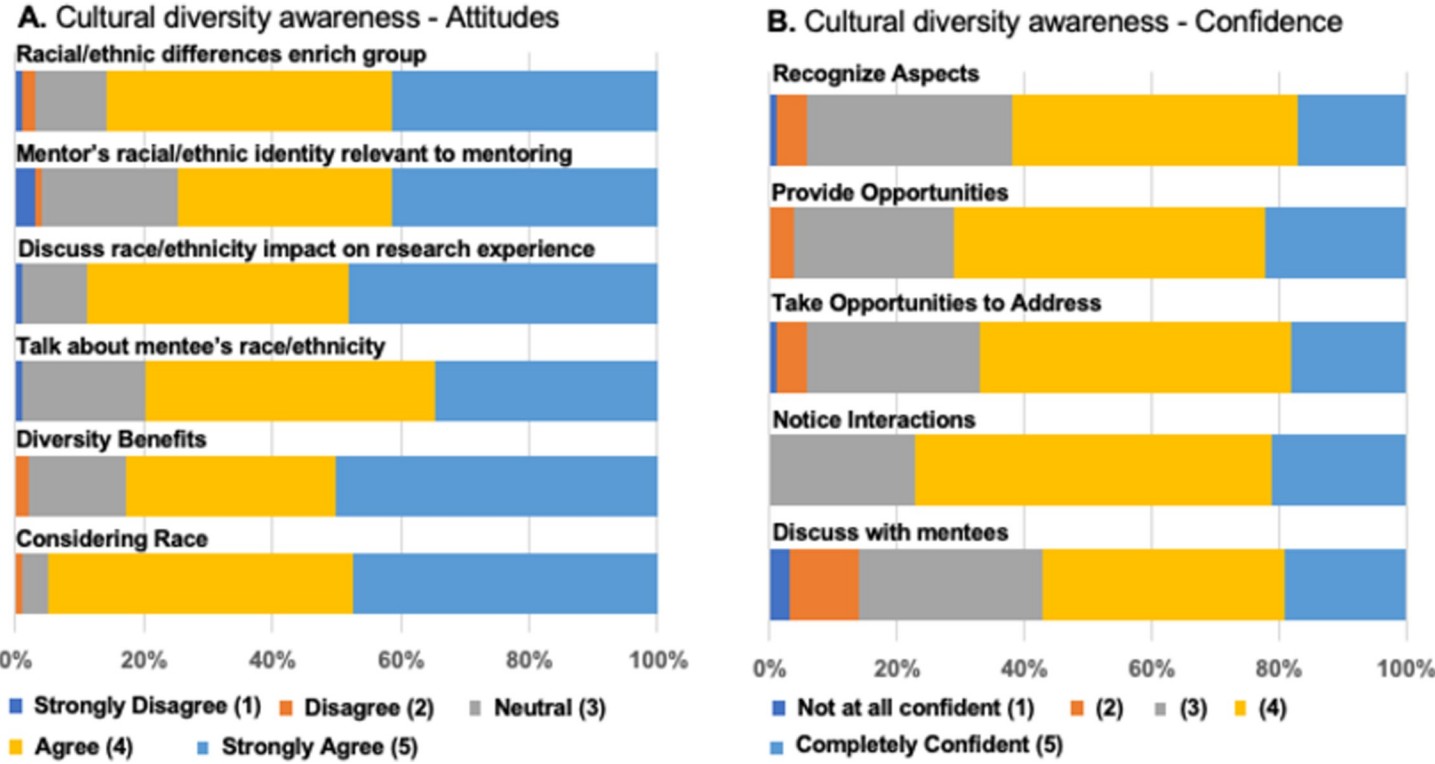

**Fig 2. Cultural diversity awareness and confidence.** A. Self-reported attitudes towards cultural diversity awareness (CDA) from all six cohorts following training. Respondents rated their agreement on a Likert-type scale ranging from 1 (Strongly disagree) to 5 (Strongly agree) to the following statements: "Racial/ethnic differences between mentors and mentees enrich the research mentoring group" [Racial/ethnic differences enrich group]; "My racial/ethnic identity is relevant to my research mentoring relationships" [Mentor's racial/ethnic identity relevant to mentoring]; "It is important for mentors and mentees to discuss how race/ethnicity impacts the mentee's research experience" [Discuss race/ethnicity impact on research experience]; "It is important for mentors and mentees to talk together about the mentee's racial/ethnic relationships" [Talk about mentee's race/ethnicity]; "Mentoring someone with a different racial/ethnic background benefits the research" [Diversity benefits]; "It is important to consider the mentee's and mentor's race/ethnicity in the mentoring relationships" [Considering race]. B. Self-reported confidence to enact various behaviors reflecting CDA from all six cohorts following training. Respondents rated their confidence on a Likert-type scale ranging from 1 (not at all confident) to 5 (completely confident) on the following skills: "Recognize aspects of the research mentoring experience that may make racial/ethnic minority students feel vulnerable to confirming stereotypes" [Recognize aspects]; "Provide opportunities for mentees to talk about their racial/ethnic identity as it relates to their research experience should the occasion arise" [Provide opportunities]; "Take advantage of opportunities to address race/ethnicity in the research mentoring relationship" [Take opportunities to address]; "Notice interactions in the mentoring relationship that could be insulting or dismissive of mentees because of their race/ethnicity" [Notice interactions]; "Discuss with mentees how it feels to be a minority in science" [Discuss with mentees].

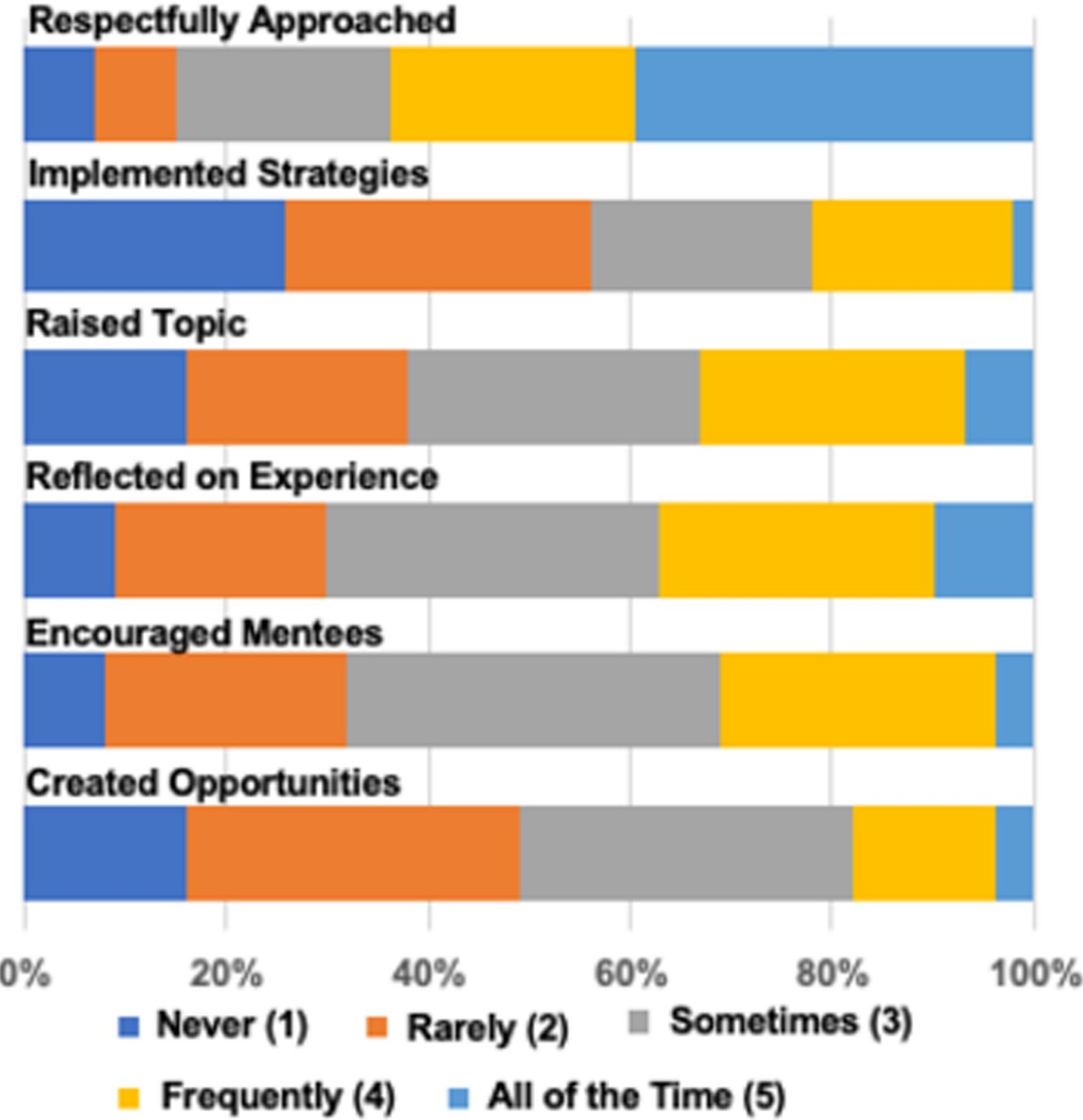

**Fig 3. Self-reported frequency of culturally aware mentorship behaviors.** Mentors from all cohorts were asked to rate the frequency with which behaviors occurred in their mentoring relationships in the last year. Answers were on a Likert-type scale ranging from 1 (never) to 5 (all of the time). Behaviors included the following: "I approached the topic of race/ethnicity with my mentee(s) in a respectful manner" [Respectfully approached]; "I implemented specific strategies to address racial/ethnic diversity in my research mentoring relationship" [Implemented strategies]; "I raised the topic of race/ethnicity in my research mentoring relationships when it was relevant" [Raised topic]; "I reflected on how the research experience may differ for mentees from different racial/ethnic groups" [Reflected on experience]; "I encouraged mentees to talk about how the research relates to their own lived experience" [Encouraged mentees]; "I intentionally created opportunities for my mentees to bring up issues of race/ethnicity as they arose" [Created opportunities].

mentoring relationship that could be insulting or dismissive to mentees because of their race/ethnicity," for which most faculty rated themselves as confident or completely confident (77, 77%). When asked a follow up question about their confidence in mentoring someone of a different race and ethnicity, however (n = 100), they reported high levels of confidence with most faculty rating themselves either 5 (31, 31%) or 6 (41, 41%) on a scale of 1 to 7, where 1 denotes "very low" confidence and 7 "very high" confidence.

When asked to report the degree to which they implemented culturally responsive mentoring behaviors in the past 12 months, participants recorded answers ranging from "Never (1)" to "All of the time (5)," though most participants responded somewhere between the two extremes. Among the items, "I approached the topic of race/ethnicity with my mentee(s) in a respectful manner" had the highest percentage (63, 64%) of survey respondents indicating that they did this "frequently," or "all the time." The item "I implemented specific strategies to address racial/ethnic diversity in my research mentoring relationship," had the lowest percentage of faculty (22, 22%) indicating that they did this "frequently" or "all of the time."

Bivariate correlations between the three CDA scales revealed significantly positive relationships among the variables: Attitudes and Behaviors (r = .33); Attitudes and Confidence (r = .41); Behaviors and Confidence (r = .44). The small to medium effect sizes indicate that although the variables are interrelated, there are unique aspects of how faculty mentors perceive and engage in CDA in their mentoring relationships.

## Workshop satisfaction and outcomes

After the workshop, participants were asked to rate their satisfaction with aspects of the workshop. In these surveys, 89 (93%) agreed or strongly agreed that the materials were useful, and 94 (95%) agreed or strongly agreed that the workshop was well-organized and easy to follow. Ninety-eight (99%) respondents agreed or strongly agreed that the speakers were knowledgeable, as well as indicated that the workshop was delivered in a respectful and sensitive manner. The lowest satisfaction rating had to do with time, with fifteen participants (12%) either disagreeing or somewhat agreeing that sufficient time was dedicated to the workshop. Participants also reported high satisfaction with the different modules covered in the training. These ratings did not demonstrate significant differences among modules or cohorts. Perhaps most importantly, ninety-four (97%) survey respondents indicated that they were likely or very likely to make changes to their current mentoring practices based on something they learned at the workshop. Similarly, ninety-two (98%) respondents indicated that they would recommend the workshop to a colleague.

## Student perceptions of mentoring

The number of students who met study criteria for each year in a Vanderbilt Basic Sciences doctoral program are as follows: 2015 (6th year students), n = 3; 2016 (5th year students), n = 15; 2017 (4th year students), n = 6; 2018 (3rd year students), n = 2. We invited all of these students to participate with the goal of recruiting six to ten participants [32] and yielded eight. Student participation was spread across years of the program, with the most students coming from years four and five. We do not report participant race and gender in an effort to protect the confidentiality of those who participated, although all students were IMSD program participants, and thus come from PEER groups. Thus, our focus groups consisted of eight 3rd–6th year IMSD students whose mentors had attended the mentorship education workshop during the time the student was working in their labs.

Overall, the students we interviewed were happy with the mentoring they were receiving at Vanderbilt. They mentioned previous positive mentorship experiences (e.g., in undergraduate

programs) and the ability to change mentors once at Vanderbilt as experiences and opportunities that helped them identify what they wanted in a mentor and develop strategies for receiving it. Students also noted being satisfied with diversity among doctoral students at Vanderbilt, but disappointed in the lack of faculty diversity. One student noted,

One of the nice things about the IMSD program is that you know 30% of your peers will be people of color, so you know there is diversity coming in. But there isn't much diversity in the faculty. . .it is challenging to go into academia. . .or industry or wherever. . .when you don't see faculty and leadership that looks like you. (Focus group 1, Participant 2)

Alongside lack of faculty diversity, students noted varied willingness and capabilities among faculty when it came to mentoring PEER students, and using strategies known to be aligned with culturally responsive mentoring. Students' responses about their mentors centered on themes related to faculty creating opportunities to discuss race, helping students connect research with lived experience, and observations about institutional structures and policies that incentivize good mentoring.

**Creating opportunities to discuss race.** When students were asked if their mentors were comfortable with and willing to bring up race, most students reported that they were. That said, students emphasized that *how* mentors brought up race and talked about it mattered just as much as their willingness to do so.

So, my mentor also talks about race quite a bit. . .especially within the past year. And he's somebody who he feels like in every aspect of life, his duty is to mentor and he wants to give advice. . .and the race thing is not one where I want to talk to him about it because he doesn't know anything about it, and I think he wants to understand and he's trying to be empathetic, but I have no interest to talk to him about that at all. So I kind of resent every time he brings it up. (Focus group 2, Participant 3)

This viewpoint was reinforced by another participant.

Yeah, I've had that same exact experience with my mentor. Whenever a topic of race or diversity would come up, he would love to talk about how he's first generation and so that makes him diverse, and basically try to equate our situations. . .I'm like that's not the same thing. (Focus group 2, Participant 4)

Other students spoke up and contrasted these experiences with mentoring that they identified as more positive. A difference seemed to be that mentors whose willingness to raise issues of race was perceived as helpful when people raised them in the spirit of making space for discussion rather than sharing opinions or comparisons, as the following exchange demonstrates:

My mentor is one of those individuals where she's aware of these things that are going on, like one particular example is when the George Floyd verdict came out, she understood (the significance) of what was going on and asked, "how are you feeling about everything?" She allows me to be open with her and have that open conversation, so I definitely think she's pretty good at having those certain conversations. I guess she identifies as a white woman and she knows she only knows things to a certain extent. . .but I think that she's willing to have an open ear and to listen and, you know - I'm here for you if you need anything. . .I feel okay having those sorts of conversations with her - it never feels too awkward. (Focus group 2, Participant 6)

Similarly, another student said,

Yeah, my mentor is also open to talking about stuff to do with race, we actually - last year with when the protests happened - we had a lab conversation about race and she made it very clear that if we ever want to talk about something with her or we feel like we need to take a day off to deal with what's going on in the news, or whatever that we can do that. So it's not brought up very often but she made it very clear that it's okay to talk about and I think everyone was comfortable talking about that stuff with her. (Focus group 2, Participant 8)

**Connecting research with lived experience.** In terms of mentors helping students connect their science to their personal lives, an important aspect was acknowledging the validity and value of educational and professional goals that were different than their mentor might have pursued. These goals included living in a particular area of the country to be close to family members (even if that limited a student's job prospects) and being responsive to student desires to work outside of academia or an R1 university environment. Describing a positive experience with their mentor, one student said,

> So, for example, I was writing a grant, and as part of that we have our career statements. And I've always said one of the things I want to do when I finish my PhD is to go back and teach at an historically black college, because that's where I went to school. My mentor has never downplayed that. . .she's never pushed me to go R1. She's always like 'okay, you know the stuff that you want to do.' This has allowed me - as much as I want to - it allows me to kind of bring myself into my work and my goals. (Focus group 2, Participant 6)

This theme emerged again in the later focus group about what a person would want from an ideal mentoring relationship; how someone would want to change their experience with mentoring:

> I think the first thing would be to listen to what I want about my future goals more. So I think my PI, kind of just assumes everyone wants to be a PI at an R1 like he is, and he doesn't really change his training based off of his trainees goals. (Focus group 3, Participant 4)

In addition to wanting mentors who recognized and appreciated the validity of different professional goals, students appreciated when mentors encouraged them to bring their "whole selves" into their work. Although some students were clear that it was important to them for their personal and professional lives to be separate, other students wanted mentors who knew them personally and were willing to work with them around their individual strengths, limitations, and passions. In an exchange between three students, participants noted the importance of having mentors who would work with them when they needed extra support, time off, or flexibility in terms of work requirements:

> PI has been pretty amazing, as to letting me do things when I can do them. . .She's been remarkably flexible with what's going on in my own life and allowing me to work through my PhD on my own terms, and my committee has been pretty fantastic about that too. And so, even though I've been sick. It hasn't like slowed down my progress, so that's been great. (Focus group 2, Participant 5)

> I've had a similar experience. (Focus group 2, Participant 7)

> Yeah, I had a personal issue last fall, and I felt comfortable going to my mentor talking about it, and I was just in a really bad place, and I didn't go into work for a week and she was fine with that. And, and I felt really supported, knowing that, and she's just really supportive in general. (Focus group 2, Participant 8)

For other students, bringing one's "whole self" meant having a mentor who recognized their strengths and passions outside of the lab. One student described how her mentor gave her a platform for pursuing her passion for supporting women and minorities in science. She said,

So I have a lot of passion for minorities and women in science and my PI could definitely see that. And when the documentary Picture a Scientist came out, which talks about harassment against women in science, [my PI] asked me to co-facilitate a lab meeting about the documentary with my lab, which I was so excited about it. . .I think a mentor that can see your passion for anything - even if it's not data driven at the bench doing experiments - helps to see the entire scientist as a human being. . .(Focus group 3, Participant 1)

**Institutional incentives for mentoring.**   At the end of the focus groups, we asked if students had anything else they wanted to share with us about mentoring. This opened an interesting and insightful conversation about if and how the university incentivizes mentoring and discourages racism among faculty. What was most important about this conversation is the insight the students had about problems of mentoring and discrimination of various types (racism, sexism) being structural problems within the university as opposed to strictly being about shortcomings of individual people and their ability (or lack thereof) to mentor underrepresented students. For example, students suggested that the University and/or Department should collect data on student experiences with particular mentors, and use that information when students are choosing faculty mentors.

I don't know if it's possible to do this, but it seems like it would be good to look at the history of the PI as a mentor before pairing (them with students). . .I mean, I know we pick our mentors, but if there was a committee that looked at the relationship between mentors and especially diverse students to avoid problems in the future. . .I don't know. It seems like that would be beneficial. (Focus group 3, Participant 4)

Another, similar suggestion had to do with working with faculty in a more targeted way to improve their mentoring:

I guess also, going off of what others are saying in terms of feedback for professors. I currently don't know if there is a mechanism in place where mentors get feedback - I don't know if it would be from the chair of the department or the dean–but in terms of if they do see patterns. (For example, the chair or dean saying to the faculty member) 'You've had four people leave your lab in the last three years. Are there issues that we need to address or that we can change?' and then provide targeted training or something like that. (Focus group 3, Student 3)

This student went on to say,

I know a lot of the R1 faculty say that 'research is our top priority. Everything else comes secondary. If it's not for a grant or for a paper, I don't have time for it.' So another question is how do we incentivize or change the perspective (of the mentors) so that the students who are creating this data for your grants and publications, they need to have high quality mentorship from you as well. (Focus group 3, Participant 3)

A summary of student recommendations is included in Table 4.

## Discussion

Mentorship education interwoven with dialogue on cultural awareness was introduced by the Vanderbilt Basic Sciences to improve the research training environment for doctoral students

from PEER groups. In this mixed methods case study, we found evidence that faculty mentors value culturally responsive mentoring beliefs and practices. At the same time, faculty reported somewhat less confidence with implementing these practices, and almost all indicated that they intended to make changes to their mentoring after the workshop. Faculty reported a variety of reasons for participating in the workshop, and few faculty reported previous participation in any kind of formal mentoring training. Overall, the PEER doctoral mentees who participated in the focus groups noticed and valued the efforts of their mentors to be culturally aware in their mentoring relationships. We highlight several findings and their implications for systemic efforts to diversify STEM, referencing components of the Hurtado et al. [20] organizational learning (OL) model for advancing inclusive science where relevant.

First, our findings reveal that faculty have varying motivations for participating in mentorship education, especially focused on increasing their cultural awareness in mentorship. Similar to research from Butz et al. [26], although many faculty in our study were intrinsically motivated to participate in mentorship education, some did so because of their faculty colleagues' encouragement. This highlights the role of positive peer influence in increasing faculty buy-in to engage in mentorship education. Byars-Winston et al. [33], found that culturally aware-focused mentorship education promoted behavioral changes not only in faculty mentoring practices, but also in their relationships with faculty colleagues and administrators. Following Hurtado et al.'s [20] OL model, mentorship education is an important factor to create faculty buy-in to gain new knowledge about PEER trainees that change faculty mindsets and mentorship behaviors.

Second, our findings revealed a value-action/will-skill gap for faculty mentors in their cultural diversity awareness. Specifically, where many reported high endorsement of the importance of culturally diversity awareness (CDA) in their mentorship, they reported comparatively lower confidence to enact behaviors consistent with that awareness. This finding is consistent with House et al. [34], who also found that faculty mentors reported challenges in their self-efficacy to enact actual culturally aware mentoring behaviors. Findings from our two-day workshop can raise faculty awareness and provide strategies for advancing culturally responsive mentorship practices. But institutions would do well to leverage such interventions to build opportunities for continued learning and practice, consistent with Hurtado et al.'s [20] OL model for changing faculty mindsets and behaviors. Ongoing mentorship skill-building can be supported by creating opportunities for faculty to share and disseminate knowledge and skills related to effective mentorship, like communities of practice.

Third, student responses demonstrated that faculty were implementing culturally responsive mentoring practices, and–for the most part–they were appreciative of these efforts. Although Vanderbilt mentor education workshops began in 2017, it is worth noting that the focus groups were conducted in the spring of 2021, a time in which structural racism,

**Table 4. Student recommendations for mentors.**

| Level of change | Recommendation |
|---|---|
| Interpersonal | • Approach conversations about race and ethnicity by creating space for dialogue, rather than giving advice<br>• Appreciate and value different professional trajectories, even if they are different from your own<br>• Facilitate opportunities for mentees to lead from their talents and passions while in your lab<br>• Understand and acknowledge that good mentorship is an integral part of generating good science (not something that detracts from it) |
| Institutional | • Develop a mechanism for students to provide anonymous feedback for mentors<br>• Leadership should use feedback to encourage improvement in mentoring when needed<br>• Culture shift that communicates value of mentoring and incentivizes it |

discrimination, and violence were at the forefront of interpersonal and institutional conversations. This context undoubtedly affected mentor/mentee relationships, as illustrated by student quotes about faculty providing space to discuss and process the death of George Floyd. It is also clear from student quotations that faculty were broaching the topic of race before the 2020 racial justice movements in both helpful and unhelpful ways.

Students noted that some faculty efforts landed better than others, and that some mentors were more skilled in culturally responsive mentoring practices than other mentors. For example, most students noted that their mentors made space to discuss race when relevant and listened to the students' experiences. The students were also clear that *how* mentors broach the subject of race was critical to whether it is helpful or hurtful. When non-PEER faculty raised the topic of race and approached it from a position of expertise, this was received as off-putting by students. Although it goes without saying that a white mentor conveying expertise about race to a PEER student is likely to be unhelpful, this points to an expertise paradox for some faculty. Whereas faculty mentors hold scientific expertise–and thus are rightly leaders in that realm, students hold expertise about their life experiences and needs that should be regarded as equally important in their professional training. Nathan, Koedinger, and Alibali proposed the concept of expert blind spot to capture the overgeneralization of one's expertise in a given domain to another [35]. Relative to culturally responsive mentorship, the experiential expertise of PEER students may eclipse the disciplinary expertise of faculty mentors, thus disrupting well established hierarchies with respect to professional rank (which is earned) and race (which is not).

The notion of cultural humility may be useful to faculty mentors enacting culturally responsive mentorship practices. Cultural humility [36] is an interpersonal stance that is other-oriented rather than self-focused and involves a willingness to reflect on oneself as a cultural being and openness to hearing and understanding others' cultural backgrounds and social identities. Thus, culturally humble faculty mentors display a respectful curiosity for others' cultural identities, resist the human tendency to view their personal beliefs and values as superior, are mindful that their understanding of others' cultural backgrounds is limited, promote a culturally self-aware mindset, and view the practice of cultural humility as a lifelong learning process [37].

Finally, students recommended that Vanderbilt as an institution develop policies that relay the value of mentorship. This recommendation came without specific prompting, mostly offered in response to the final question of the focus group, "What else do you want me to know?". This is noteworthy considering that mentorship is often considered a practice that operates and creates change at the interpersonal level. Students, however, seemed to see the need for institutional-level policies to ensure effective mentorship, recognizing a significant opportunity for improved mentorship to lead to shifts in the institutional climate. The students' recommendation emphasizes the positive impact of social norming on faculty behaviors that departments and training programs can communicate: "formal mentorship education and effective mentorship are valued and expected here." Indeed, Vanderbilt Basic Sciences has made some progress in this regard, now requiring faculty to participate in an in-person mentorship education session in order to accept doctoral students into their laboratory.

In addition to improving resources for faculty, students suggested the need for faculty to receive feedback on their mentoring, as they do for other job functions–e.g., peer review for research and course evaluations for teaching. In this scenario, if faculty mentors repeatedly struggled to provide positive mentorship experiences for their trainees, students wanted the institution to respond by addressing the issues with individual faculty members. There are challenges to this, of course. Given the power imbalance between students and faculty, student feedback, even when constructively given, can place the student at some risk. The opportunity for retaliation can never be truly eliminated, both during the student's time in training and

further on in their career. The subjective nature of individual mentee-mentor relationships makes them difficult to evaluate–is this a poor mentor or a poor match? Repeated poor outcomes may very well be addressed in a faculty performance review, but such reviews are rightly confidential. Participant 3 in the third focus group was absolutely correct that faculty are often under considerable pressure to obtain grants and publish papers in an extremely competitive environment. The harmful toll that this pressure takes on quality mentorship for students deserves greater recognition. Changing social norms is a critical part in Hurtado et al.'s [20] OL model. The theory of planned behavior [38] suggests that injunctive norms, or the expected behaviors in a given social context - like a department - will become reinforced in that context. Change the institutional expectations, and people will feel more compelled to follow them.

## Limitations

Although we complied with best practices in case study methods such as employing a well-bounded case, triangulating data sources and data types, and working iteratively through data analysis, interpretation, and author reflection, there are nevertheless limitations. Survey response rate was 64% across all cohorts for survey data, and 8 out of 26 invited students elected to participate in the series of three focus groups. We do not know whether the non-responses reflect lack of time, lack of interest, or something else; nor do we know whether non-responders would have shared similar perspectives to those who participated. We also did not expect all students to be aware of if and when their faculty mentors had participated in the mentor education workshop, therefore, we did not ask them directly about the training. Furthermore, because our case study does not follow individuals across time, we did not assess individual change with respect to mentoring behaviors or the durability of mentors' intentions to change. Finally, we did not conduct within-group data analyses nor use an intersectional lens with respect to mentor or student experiences. These experiences *are* represented in the qualitative data to some extent, however, we did not probe for them. As a result, we have missed some of the ways in which gender, class, sexuality, and other identity categories intersect with race to shape mentor-mentee relationships and their potential for driving institutional change, a clear direction for future research.

## Conclusion

There are promising approaches to helping faculty becoming more culturally aware and to improve their skills in culturally aware mentoring practices [10, 16, 17, 33, 34]. There are several implications from our study. One is the power of locally harvested data about the experiences of PEER doctoral students that can serve to raise faculty awareness of needed change, help to identify and guide interventions, and even be a motivating factor for faculty participation in those interventions. Another implication is the importance of well-coordinated, multi-prong institutional-level interventions to affect organizational learning needed for systemic change. For example, prior to the mentorship education intervention described in this study, Vanderbilt Basic Sciences had adopted holistic admissions processes that no longer required the Graduate Record Examination and had conducted anonymous student evaluations of their mentors at exit after completing their doctoral degree. From those evaluations, they were able to document a positive correlation between students' mentorship satisfaction and time to degree completion [8], and this finding became another motivating factor for faculty investment in effective mentorship. Hurtado et al.'s [20] work underscores the important role of buy-in from STEM faculty to value culturally responsive practice and inclusive environments. They are important change agents who have the influence and agency to advocate for and enact transformational practices and policies that enhance the success of PEER students in the

biomedical sciences. Findings from this present study demonstrate that evidence-based mentorship education can be part of a strategy to catalyze institutional change toward culturally responsive research training environments.

## Supporting information

**S1 File. Study measures.**
(DOCX)

## Acknowledgments

This study was supported by the Office of the Dean of Basic Sciences, Vanderbilt School of Medicine; the Center for the Improvement of Mentored Experiences in Research (CIMER) in the Wisconsin Center for Education Research at the University of Wisconsin-Madison; and the Department of Medicine at the University of Wisconsin-Madison. We would like to acknowledge the staff of the Vanderbilt Office of the Dean of Basic Sciences, especially Tracy O'Brien, Becky Sanders, Steven Doster, and Anne Lara, for their assistance with workshops, and the unwavering support of Larry Marnett, Vanderbilt Basic Sciences (VBS) Dean Emeritus and Jennifer Pietenpol, Chief Scientific and Strategy Officer, VUMC. We thank the Vanderbilt Office of Biomedical Research Education and Training (BRET), in particular Roger Chalkley and Abigail Brown, for assistance with the doctoral student Climate, Culture and Career Plans survey, as well as Felysha Jenkins, VBS Assistant Dean for Diversity, Equity and Inclusion and Larry Marnett for helpful comments on the manuscript. Finally, we thank the CIMER Evaluation and Research team for their assistance with mentorship education evaluation data collection.

## Author Contributions

**Conceptualization:** Sarah Suiter, Angela Byars-Winston, Christine Pfund.

**Data curation:** Fátima Sanchieznieto.

**Formal analysis:** Fátima Sanchieznieto.

**Funding acquisition:** Linda Sealy.

**Investigation:** Sarah Suiter, Angela Byars-Winston, Christine Pfund, Linda Sealy.

**Methodology:** Sarah Suiter, Angela Byars-Winston, Fátima Sanchieznieto, Christine Pfund.

**Project administration:** Sarah Suiter, Linda Sealy.

**Supervision:** Linda Sealy.

**Writing – original draft:** Sarah Suiter, Angela Byars-Winston, Fátima Sanchieznieto, Christine Pfund, Linda Sealy.

**Writing – review & editing:** Sarah Suiter, Angela Byars-Winston, Fátima Sanchieznieto, Linda Sealy.

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
