## [Decision Letter · Decision Letter 0]

30 Oct 2023

PONE-D-23-26718Utilizing Mentorship Education to Promote a Culturally Responsive Research Training Environment in the Biomedical SciencesPLOS ONE

Dear Dr. Suiter,

Thank you for submitting your manuscript to PLOS ONE. After careful consideration, we feel that it has merit but does not fully meet PLOS ONE’s publication criteria as it currently stands. Therefore, we invite you to submit a revised version of the manuscript that addresses the points raised during the review process.

In addition to the reviewer's comments, there are additional points that I suggest you to address in your manuscript revision:Line 224: Please ensure that the word "methods" is in the plural form, as it should be "methods."In the initial mention of the results, it would be helpful to state the total number of participants and then provide the percentages.Please provide more information about the focus group involving doctoral students. Specifically, clarify how many participated.It is important to include a section discussing the limitations of your study and discussion of potential sources of bias.

We look forward to receiving your revised manuscript.

Kind regards,

Delfina Fernandes Hlashwayo, Ph.D.

Academic Editor

PLOS ONE

Journal Requirements:

Reviewers' comments:

Reviewer's Responses to Questions

**Comments to the Author**

1. Is the manuscript technically sound, and do the data support the conclusions?

Reviewer #1: Partly

Reviewer #2: Partly

Reviewer #3: Yes

2. Has the statistical analysis been performed appropriately and rigorously? 

Reviewer #1: No

Reviewer #2: Yes

Reviewer #3: Yes

3. Have the authors made all data underlying the findings in their manuscript fully available?

Reviewer #1: Yes

Reviewer #2: No

Reviewer #3: Yes

4. Is the manuscript presented in an intelligible fashion and written in standard English?

Reviewer #1: Yes

Reviewer #2: Yes

Reviewer #3: Yes

5. Review Comments to the Author

Reviewer #1: The authors of “Utilizing Mentorship Education to Promote a Culturally Responsive Research Training Environment in Biomedical Science” present a case study of work conducted over several years at Vanderbilt University. The authors use a mixed methods approach to describe the represented workshops to train faculty to support the diversification of the scientific workforce.

The program, intervention, and data descriptions are worthwhile to present. These data showed that faculty had high post-survey scores, including satisfaction and confidence in cultural diversity awareness. Qualitative data demonstrates key anecdotes, which would be essential to report associated with their data descriptions.

Surveys associated with the data have been made available and could be linked in the article or added as supplementary materials, consistent with an open data policy. The manuscript will written. However, some sections could use some clarity to improve their readability.

Of concern, the current data only partly support the claim that the intervention impacted the faculty. The qualitative data present is most compellingly presented. However, the quantitative data do not support this claim. The authors could address their stated claims by linking the presented post-survey data to the pre-survey results. The authors should describe the assessment framework they used in the analysis. The authors could remake the figures to present the data more clearly and completely describe any statistical analyses in the qualitative data section, if any. Finally, the authors could move information from the introduction to the discussion and expound on the methods to support the readability and clarity of the presented research.

Reviewer #2: Thank you for the opportunity to review this manuscript. The manuscript provides an overview of a culturally responsive mentorship education program offered at Vanderbilt University. The manuscript is presented as a case study, yet this presentation is confusing and difficult to follow. I think the manuscript could benefit from being reworked as a traditional manuscript with introduction, methods, results, and discussion format. There is a lot of information in the Introduction and Case Study sections that could be used in the discussion to better put this study into context.

1) The introduction and Mentorship Education as a Systemic Intervention sections are repetitive – the way it reads is like there are two introductions. These sections could be pared down and streamlined.

2) I think the paragraph in lines 120 to 130 could be moved to the discussion. It’s confusing to have it presented here because you haven’t introduced anything about the data collection yet.

3) Is the goal of training the mentors this way to promote inclusion?

4) Do you have any measure of how many mentors were invited to attend these workshops or how many mentees these mentors collectively have?

5) You state that this study has 3 aims, the third of which is to see how students perceived faculty efforts at enacting culturally responsive mentoring. However, the questions posed in the focus groups don’t directly address this. The methods indicate that the focus groups focused on participants mentoring expectations, mentoring experiences, and changes to mentoring, which don’t address changes or perceptions regarding culturally responsive mentoring.

6) Information on participants in the focus groups is split up under two different headings making it difficult to understand the full picture.

7) Were continuous variables normally distributed? Is that why means and standard errors were calculated?

8) On line 352, the authors indicate that participants were reluctant. However, to me it seems to indicate external influence as a motivator more than reluctance.

9) line 448 – what kind of bivariate correlations? This should be in methods.

10) The introduction largely focused on the importance of belonging and inclusion for PEER. However, this did not appear to be a theme of the qualitative findings. Why do you think so?

11) Line 629 states that the education had a positive impact on mentors self-ratings for quality of their mentorship relationships and mentoring effectiveness. However, there is no indication that this is correct in this dataset since there is no control group or pre-education assessment with which to compare. Similarly, I didn’t see a lot of findings in the qualitative analysis that indicated that mentees noticed the improved efforts of their mentors because of cultural awareness training.

12) There are a number of statements in the Case Study section that could benefit from further explanation. For example, the sentence on lines 174 to 176 “Certainly, there are few faculty from PEER groups, since faculty diversity in the Vanderbilt School of Medicine is considerably lower than graduate student diversity” had me questioning how much lower. I think you answer this question later on, but why introduce it here incompletely? Also, you mention on line 178 that inclusion has lagged. How do you know that inclusion has lagged? Have you measured it?

13) Minor- please spell out NASEM the first time it is used.

Reviewer #3: Overall, this manuscript reports on a critical topic. The case study is well-executed. I do not have specific points for the authors, but rather one global one that I think seriously detracts from the manuscript. That is, the manuscript is exceedingly long. The level of detail is way too much, which makes it difficult to read. I found myself when reading that I would skim over sections that were not informative. For example, the reader has to read 9 pages before the methods section. The text repeats what is in the tables. All of the items for each of the measures are included in the text. Many of the quotes from the qualitative portion are entirely too long and there are too many of them. The presentation of the qualitative data extends over 7 pages. The discussion section is over 4 pages long. With such lengthy sections and the granular level of detail provided, the reader loses sight of the purpose of the manuscript and the take home results.

6. PLOS authors have the option to publish the peer review history of their article (what does this mean?). If published, this will include your full peer review and any attached files.

Reviewer #1: **Yes: **Jose A. Rodrigues

Reviewer #2: No

Reviewer #3: No

---

## [Author Response · Author response to Decision Letter 0]

14 Jan 2024

Sarah V. Suiter, PhD, MS

Human & Organizational Development

Vanderbilt University

230 Appleton Place Peabody #90

Nashville, TN 37203

January 14, 2024

Dear Editors,

My co-authors and I are resubmitting our paper, “Utilizing Mentorship Education to Promote a Culturally Responsive Research Training Environment in the Biomedical Sciences”for your review and consideration for publication in PLOS One. We are grateful to the reviewers and appreciate their thorough and helpful feedback. We have taken this feedback seriously and responded to each point in the reviews. Below, you will find a list of the reviewer’s suggestions followed by the changes we have made. In our document uploads, we have included a “clean” version of the edited manuscript, as well as a document with track changes. Please note that we made our formatting, reference, and a few minor editorial changes to the “clean” document after we accepted track changes, so those changes are not tracked in the track changes document; our substantive edits nevertheless are. 

Responses to Editors’ Requests:

● Line 224: Please ensure that the word "methods" is in the plural form, as it should be "methods."

o We have changed Method to Methods

● In the initial mention of the results, it would be helpful to state the total number of participants and then provide the percentages.

o Both in the tables and in the results text, we have included both percentages and numbers. Because the number total number of survey respondents is 108, n’s and percentages are very similar. We also note that some questions will have fewer than 108 respondents. These have been marked where appropriate. We leave it up to the editor’s discretion whether to keep just %’s, numbers, or both

● Please provide more information about the focus group involving doctoral students. Specifically, clarify how many participated.

o We have clarified how many doctoral students participated in the focus groups in the section on “Data Collection and Participants”

● It is important to include a section discussing the limitations of your study and discussion of potential sources of bias.

o Thank you for this suggestion. We included a “Limitations” section between the Discussion and Conclusion sections of our original paper. We have edited the section in our resubmission to address specific issues raised by the editor and reviewers.

● Please ensure that your manuscript meets PLOS ONE's style requirements, including those for file naming.

o We have ensured that our manuscript meets PLOS ONE’s style requirements

● In your revised cover letter, please address whether your data are prohibited from being made public for ethics reasons or if they can be accessed in a data repository

o De-identified quantitative data from participant surveys has been made accessible via Zenodo at [https://zenodo.org/doi/10.5281/zenodo.10372674]. Qualitative data from pre-participation surveys and from interview transcripts will not be publicly available to protect participants from potential identification.

● Please include your full ethics statement in the ‘Methods’ section of your manuscript file. In your statement, please include the full name of the IRB or ethics committee who approved or waived your study, as well as whether or not you obtained informed written or verbal consent. If consent was waived for your study, please include this information in your statement as well. 

o We have included our full ethics statement in the first paragraph of our Methods section

Reviewer #1 Comments with Responses:

● The quantitative data currently do not support the major claims in this article. The pre and post-survey must be linked, and the difference in Likert scales after the intervention must be assessed and reported with corrected p values. The current data do not support the discussion or abstract of the paper, which states, Line 629: “.. the education also had a positive impact on the mentors’ self-ratings of the quality of their mentoring relationships and their mentoring effectiveness.” OR Line: 35 “...increased faculty mentors’ self-reported mentorship competencies.” If the pre-survey can’t be linked, then I would recommend not mentioning or discussing the impact of the intervention in another version of the manuscript.

o Thank you for this suggestion. We are unable to link pre- and post- tests for all participants, thus have removed any verbiage that would suggest change over time (e.g., impact, effectiveness, increased)

● Significant shortening is necessary for redundant verbiage throughout the article. The manuscript needs a succinct and precise description of the workshops and strategies implemented in the introduction and expounded specificity in the article's methods, particularly addressing an assessment framework used to evaluate the program and more information about the quantitative analyses.

o We have shortened the article from 35 pages (excluding figures and references) to 30 pages (excluding figures and references), provided a succinct description of the workshop in the introduction, and provided more specificity on methods, including an assessment framework.

● The authors should place the contextualization of the previous work at Vanderbilt in the discussion section of the article

o We have removed some of the contextualization text from the manuscript all together in an effort to shorten the overall length. We have retained some of the text between the Introduction and Methods section (in the “Case Study” section) because we think understanding the context of the study and what else was going on at Vanderbilt at the time of the mentoring workshops is important for helping the reader situate and interpret the findings we present. 

● A figure (flow-chart) of the participants, i.e., the total number of faculty and those who answered the assessment (pre and post-survey), i.e., the response rate

o As stated earlier, we are unable to link the pre- and post-surveys, and are not necessarily sure that everyone who took a post-survey took a pre-survey, as participation in the surveys was voluntary and participants were allowed to take one without the other. With this in mind, we think a figure of this type is likely to be more confusing than helpful. In the text of the manuscript, we have clarified that we only used one question from the pre-survey in our analyses/results, and the rest of the questions were from the post-survey. We have clarified response rates for the post-text surveys.

● The introduction needs more specific details of the study design and questions the study addresses, detailed later in the methods, including an evaluation framework

o We have provided more specific details of the study design in the final paragraph of the discussion. See the response directly following this one regarding the evaluation framework. 

● What particular evaluation model was used to assess the program? Through the discussion, I assume an organizational learning model for advancing inclusive science - Hurtado et al. There should be a clear description of this assessment model in the methods, such as in a new program evaluation section, and introduced in the introduction and abstract of the article.

o Thank you for this suggestion - it helped us realize that we needed to clarify how we used Hurtado’s model. We did not initially set out to study the mentor education workshops as part of a Vanderbilt-specific case study. Rather, the case study emerged overtime, and the data we garnered informed local (department, workshop) specific actions before they were used for this case study. Because of this, reporting an evaluation model up front as if it informed data collection methods and questions is disingenuous. Rather, we used Hurtado’s model to help situate our analyses and their interpretations vis-a-vis institutional change literature. We have clarified this in the Methods section, and have moved Hurtado’s model from the Introduction to the Methods section. We have not added a new Program Evaluation section. 

● The quantitative data section needs to be more thoroughly explained in the methods section; for example, on Line 448, Bivariate correlations between the three CDA subscales…

o A sentence in the methods section has been included to state that Pearson’s bivariate correlations were calculated for the three CDA mean scale scores.

● Figures 1-3 should be of better quality regarding the resolution. Interpreting these averages as presented in a whiskered box plot average is difficult. It may be beneficial to show Figures 1-3 together in a stacked bar plot with the specific questions on the survey on the Y axis, the % of respondents on the X axis, and the Likert scores labeled. If the average of each Likert scale is essential to address, then a whisker plot can be added to the same figure, just as one alternative example of presenting these data.

o We have taken the reviewer’s suggestions to convert the Likert scale plots to stacked bar plots, and refrain from using means and distributions and rather report n’s and percentages of respondents in the text instead.

● I would’ve liked to see the recommendations from the trainees listed in a table and brought up in the results section. As the discussion implies, you can compare NASEM guidelines with student recommendations if they are similar.

o We have included recommendations from the mentees in at table (Table 4). When we edited our discussion, we removed the section in which NASEM guidelines were listed, so we have not included them in the table. 

● Supplementary figures should include pre and post-survey and discussion questions for the focus groups.

o We have included post-survey and focus group questions as supplementary materials. We have only reported on one question from the pre-participation survey, and have included that in the text. To save space and avoid redundancy, we only include questions and scales that are reported in the text in our supplementary figure

● A supplementary table can include an example of workshop materials.

o The curriculum is quite involved and was developed based on multiple evidence-based curricula (see lines 158 - 162). Including a table with examples of workshop materials would be incredibly long and not terribly helpful to readers. We have included a link to CIMERProject.org, where a few of these curricula are available as open access. 

● The discussion focuses on the implications of the work and vaguely justifies the statements made. It would be helpful to bring up specific data presented throughout the results to justify the points made in the discussion and the conclusion.

o We have edited the discussion to ensure that the statements are supported by data presented in the results section. 

● Please rewrite the abstract to reflect on the data appropriately represented and specific methods used, i.e., comment on the quantitative, qualitative, and assessment framework, statistical analyses, if any, and discussion implications. Specifically, removing mention of impact or increasing scores if a pre-survey is not linked. 

o We have updated the abstract to reflect the methods we used and have removed mention of impact

● The introduction briefly clarifies the culturally “responsive mentorship education program.” You could replace “explained in a later section” with a succinct program description. The program should be straightforwardly described and understood without reading the entire manuscript.

o Thank you for this suggestion. We have done this. 

● General comment: Except for the methods section, I find phrasing such as “in this paper, described in this paper, cited in this paper” redundant and unnecessary; you could consider removing these phrases

o We have removed these phrases

● General comment: When listing integers of 10 or under, please write these out, i.e., six instead of 6 on Line 338

o We have made this change

● General comment: Where % s are listed, please also list an n= value. Using n with percentage values will help the reader understand the denominator associated with each piece of data.

o We have included n values wherever there are percentages listed

● General Comment: Please be consistent with using PEERs and underrepresented-students verbiage throughout the manuscript. 

o We have worked to ensure we were consistent with PEER throughout the manuscript. 

● General Comment: For the workshop stratification and outcomes section in the results, it may be helpful to show a figure demonstrating when each workshop was full to establish your point here. Otherwise, I would move specific comments on the popularity of the workshops to the discussion section.

o We have removed the comment regarding popularity of the workshops

● General Comment: For the Figures: The questions in the Likert scale should match the figure label or figure label indicated in parentheses to make the figure more accessible to the reader. For example: “Discuss with mentee how it feels to be a minority in science” (Discussing Experiences).

o We have included matching labels for the figures in the legends

● Line 78 “noted in the opening paragraph and in other publications,” as the introduction cites previous work and citations imply “in other publications.” This phrasing is redundant and unnecessary. I would recommend simply citing the literature after “based on the lack of racial and ethnic diversity in STEM fields.”

o We have made this change

● Line 82: It is unclear what is meant by “post-workshop.” please see an above comment and clarify whether the “responsive mentorship education program” was indeed a workshop. Explicitly describing the program.

o We have worked to clarify our naming of the mentoring intervention throughout (it was indeed a workshop, and have edited throughout to indicate that it was)

● Line 83: Remove: “Where specific authors contributed uniquely to the research design or data analyses, we note this contribution with their initials.” Please note the author contributions in the authors' contribution section and are unnecessary to highlight throughout the article. 

o We have removed author initials throughout, with the exception of the description of qualitative analysis – in this section, we have retained initials because describing who did what is important to understanding our analytic process

● Line 182: See the comment above about the author's contributions

o We have removed initials throughout

● Lines 154-155: “in this paper” is a redundant comment 

o We have removed this

● Line 526: please check if the quote is correctly listed: “I think everyone goes comfortable talking about that stuff with her.” If so, use [] to add what the authors think the intended word was.

o Thank you - this word should be “was” - we have changed it in the manuscript

● Line 590: remove “-questions we didn’t ask but should have, other things that had occurred to them along the way, etc.” etc. trivializes the rest of this phrase

o We have removed this

● Line 605-620: The last quote is rather long. Can we shorten the section while providing the contextualized information needed to illustrate the point about faculty development?

o Yes - we have edited the quote to be more succinct, and broken it in to two (the same student said it, but they were presenting two different ideas - we have split the quote accordingly)

Reviewer #2 Comments with Responses:

● I think the manuscript could benefit from being reworked as a traditional manuscript with introduction, methods, results, and discussion format. 

o We have considerably shortened our Introduction and Case Study section, and our paper now follows this format, with the exception of a brief “Case Study” section after the Introduction. If the reviewers prefer, we think this section could also be named “Study Context”

● There is a lot of information in the Introduction and Case Study sections that could be used in the discussion to better put this study into context.

o We have removed information from the Introduction and Case Study and have edited the Discussion to clarify our primary takeaways with respect to the study. 

● The introduction and Mentorship Education as a Systemic Intervention sections are repetitive – the way it reads is like there are two introductions. These sections cou

---

## [Decision Letter · Decision Letter 1]

14 Feb 2024

PONE-D-23-26718R1Utilizing mentorship education to promote a culturally responsive research training environment in the biomedical sciences

PLOS ONE

Dear Dr. Suiter,

Thank you for submitting your manuscript to PLOS ONE. After careful consideration, we feel that it has merit but does not fully meet PLOS ONE’s publication criteria as it currently stands. Therefore, we invite you to submit a revised version of the manuscript that addresses the points raised during the review process.

In addition to the reviewer's comments, there are additional points that I suggest you to address in your manuscript revision:

**Abstract:** The inclusion of numerical data would significantly enhance the clarity of your abstract, allowing for a more comprehensive understanding of the methods and results in quantitative terms. Consider incorporating such data to ensure that your abstract transcends beyond the qualitative aspects, providing a well-rounded overview of your study.

**Introduction: **Clarify whether the outlined content in lines 55-56 pertains to a specific country, and if so, explicitly mention the country.

**Methods: **Ensure that the results of the study are not presented within the methods section.

**Results: **

The sentence structure in lines 336-341 is excessively long and could benefit from rephrasing to improve readability.Confirm the placement of quotation marks in lines 499-508, and other instances where quotes are utilized.

**Discussion:** Consider introducing a new subtopic in the discussion section to address the attempts made by students to highlight issues with certain mentors (as outlined in lines 529-533 and lines 536-542). Discuss the necessity of creating mechanisms to evaluate the mentor-mentee relationship, as well as the performance of mentors, emphasizing the potential implications of these dynamics.

**Conclusion:** The conclusion is notably concise. Consider expanding the conclusion to provide a more comprehensive overview. Additionally, suggest future directions or implications based on the study’s results.

**Additional comment: **Despite one reviewer suggesting the reduction and placement of quotes in supplementary materials, please consider retaining quotes in the main manuscript, particularly those related to student perceptions. These quotes contribute meaningful context and a thorough explanation of the results.

We look forward to receiving your revised manuscript.

Kind regards,

Delfina Fernandes Hlashwayo, Ph.D.

Academic Editor

PLOS ONE

Journal Requirements:

Reviewers' comments:

Reviewer's Responses to Questions

**Comments to the Author**

1. If the authors have adequately addressed your comments raised in a previous round of review and you feel that this manuscript is now acceptable for publication, you may indicate that here to bypass the “Comments to the Author” section, enter your conflict of interest statement in the “Confidential to Editor” section, and submit your "Accept" recommendation.

Reviewer #1: All comments have been addressed

Reviewer #2: (No Response)

2. Is the manuscript technically sound, and do the data support the conclusions?

Reviewer #1: Yes

Reviewer #2: Yes

3. Has the statistical analysis been performed appropriately and rigorously? 

Reviewer #1: Yes

Reviewer #2: Yes

4. Have the authors made all data underlying the findings in their manuscript fully available?

Reviewer #1: Yes

Reviewer #2: Yes

5. Is the manuscript presented in an intelligible fashion and written in standard English?

Reviewer #1: Yes

Reviewer #2: Yes

6. Review Comments to the Author

Reviewer #1: Thank you for the opportunity to review the resubmission of Utilizing mentorship education to promote a culturally responsive research training environment in the biomedical sciences. The authors have undertaken a tremendous effort to revise the manuscript and addressed the major and minor concerns presented by each reviewer. I have some additional minor revisions to comment below:

1) Line 166, is there an extra?

2) Line 257 SPSS, please be more specific and write the full version of the program, the version, etc., similarly with R, please add the version and any additional packages that were used to analyze the data

3) In lines 275-276, a comma is missing between % and n

4) Line 340, please add contextualization for the “ p = “ this should read p value = ) and include the specific test conducted within the parathesis.

I also have some additional optional comments that might support the readability and impact of the manuscript.

5) I still find it unnecessary to separate the information in the introduction from the “Case Study” section. I would remove the entire case study section from the manuscript and move the first section within the Case study section from the introduction into the discussion for contextualization. This section could be paired with 1-2 sentences to describe why Vanderbilt was a good place for this study in the introduction. The introduction is lengthy, and this contextualization is better placed in a discussion.

6) Similarly, the mentorship education section within the case study could be placed within the methods

7) To support the reduction of text, the authors could consider putting all the quotes into a supplementary table and just referencing the quotes in the supplementary table. OR potentially only keep one quote per section and reference the additional quotes in the supplementary table.

Reviewer #2: The manuscript has significantly improved from the last version but, in general, it is still too long.

1) In the introduction, the second paragraph could be deleted since it’s not directly applicable to this case study.

2) The in-depth description of the methods starting on line 91 is better suited for the Methods rather than the Introduction.

3) The aims of this paper seem more in line with describing a new mentoring education curriculum. I suggest the paper be framed as such and the project design, along with the development and implementation of it should be presented in the methods, not as separate sections. Doing so would decrease redundancy in the manuscript.

4) line 177-178 : Delete “examines mentorship education efforts in the biomedical sciences at Vanderbilt University.” This is not something you believe your study has done, but rather something it has done. This entire paragraph should be in the Discussion rather than methods.

5) Lines 208-211 including Table 2 are results, not methods, and should be moved.

6) Line 227-229 about not asking about mentor education workshop is a limitation and should be moved to that section.

7) Lines 237-242 should be moved to results.

7. PLOS authors have the option to publish the peer review history of their article (what does this mean?). If published, this will include your full peer review and any attached files.

Reviewer #1: **Yes: **Jose A. Rodrigues, D.O., Ph.D.

Reviewer #2: No

---

## [Author Response · Author response to Decision Letter 1]

1 Apr 2024

Dear Editors,

My co-authors and I are resubmitting our revised paper, “Utilizing Mentorship Education to Promote a Culturally Responsive Research Training Environment in the Biomedical Sciences” for your review and continued consideration for publication in PLOS One. Thank you for the opportunity to respond to the reviewers’ helpful feedback. Below, we list each reviewer’s suggestions followed by our response to those suggestions and any corresponding changes we have made. In our document uploads, we have included a “clean” version of the edited manuscript, as well as a document with track changes. 

Responses to Editors’ Requests:

● Abstract. Include numerical data to enhance clarity of mixed methods in the abstract.

● We added the sample size of the faculty participants and included overall percent response rates to the variables we measured in our survey. 

● Introduction. Clarify whether the outlined content in lines 55-56 pertains to a specific country and, if so, explicitly mention the country.

● We now specify the United States in lines 55-56 “Underrepresentation persists in the US scientific workforce, where 13% of the STEM workforce and 56 only 4% of faculty at research institutions identify with a minoritized racial or ethnic group [3].

● Methods. Ensure that the results of the study are not presented within the Methods section.

● We moved the sample descriptions in Lines 208-211 from the Methods to the Results section.

● Results. The sentence structure in Lines 336-341 is long and could benefit from rephrasing. Confirm placement of quotation marks in Lines 499-508 and other places where quotes are used.

● We have rewritten this sentence, dividing it into two sentences to improve readability. The revised text now appears on Lines 294-300.

● We used block indentation to denote participant quotes, not quotation marks. We intended for our use of parentheses at the end of each block quote, specifying the focus group and participant numbers, to signal to the reader the end of quoted text.

● Discussion. Consider introducing a new subtopic in the discussion section to address the attempts made by students to highlight issues with certain mentors (as outlined in lines 529-533 and lines 536-542). Discuss the necessity of creating mechanisms to evaluate the mentor-mentee relationship, as well as the performance of mentors, emphasizing the potential implications of these dynamics

● We have added a paragraph to the discussion addressing this topic. The new text now appears on Lines 593-606.

● Conclusion. Consider expanding the conclusions to provide a more comprehensive overview. Suggest future directions or implications from the study’s results.

● We have added a suggested future research direction at the end of the Limitations section given that it logically flowed from a shortcoming noted of our present study. Further, we expanded the Conclusions section to articulate implications from our study.

Reviewer #1 Comments with Responses:

● We are pleased to know that this Reviewer responded yes to Questions 1-5, finding that we adequately addressed comments raised in the first round of reviews.

● 1) Line 166 - is there an extra question mark.

● Thank you for catching that error. We removed the extra question mark.

● 2) Line 257 SPSS, please be more specific and write full version of the program, etc and similarly with R, please add the version and any additional packages used to analyze data.

● We have now specified the version of SPSS (28) and R (4) in the text.

● 3) In lines 275-276, a comma is missing.

● Thank you for catching that error. We have inserted a comma between % and n.

● 4) Line 340, please add contextualization to the “p =” and include the specific text conducted within the parenthesis.

● We have added the H statistic, degrees of freedom and italicized the “p” value to indicate the probability of statistical significance. 

● 5) Consider deleting Case Study section and providing Case Study contextualization in the Discussion section. Introduction is lengthy.

● We appreciate the Reviewer’s suggestion to further shorten the Introduction. We have chosen to retain this section so that the reader has the contextualization of the study. Following the Reviewer’s overall recommendation to shorten the Introduction, we have pared down this Case Study section (renaming it “Case Study Context”), moved it to the Methods section, and trimmed it down to one paragraph from two paragraphs, focusing on the relevance of the study site to investigate our study questions.

● 6) Move the mentorship education section to Methods.

● We have moved this section as suggested. 

● 7) Consider putting all quotes in the Supplementary Table.

● We appreciate the Reviewer’s suggestion. We have chosen to retain the quotes in the main manuscript as they give meaningful context in explaining our results. 

Reviewer #2 Comments with Responses:

● 1) In the Introduction, delete the second paragraph. 

● We deleted the second paragraph. 

● 2) Description of the methods starting on Line 91 is better suited for the Methods section.

● We agree. In keeping with the Reviewer’s suggestion here and in point #3 below, we have significantly edited the Introduction so that it does not mix study design and methods overview in this section. This edit also reduces redundant description of methods previously given in both the Introduction and in the Methods sections. 

● 3) The aims seem to be more aligned with describing a new mentoring education curriculum. Suggest that the paper be framed as such and present the development and implementation of the curriculum in the Methods.

● We appreciate the Reviewer’s sentiment that the paper seems focused on describing the development and implementation of a new intervention. As captured in our manuscript’s title, our goal is to examine the impact of a mentorship education intervention on promoting a culturally responsive research training environment. To that end, we have trimmed and/or moved some of the text earlier in the Introduction section so that the study aims are described up front. We moved descriptions of the Case Study Context and the intervention itself to the Methods section and trimmed the text in both sections. We hope that these edits de-emphasize the development and implementation of the intervention and center the focus on understanding its impact on faculty participants and their mentees.

● 4) Lines 177-178, delete “examines mentorship education…Vanderbilt University.”

● We have deleted this from the text.

● 5) Lines 208-211 including Table 2 are results, not methods. Move to relevant section. 

● We moved the sample descriptions in Lines 208-211 from the Methods to the Results section. 

● 6) Lines 227-229 regarding not asking about mentor education is a limitation and should be moved to that section.

● We moved this sentence to the Limitations section. 

● 7) Lines 237-242 should be moved to Results.

● We moved this section of text regarding student eligibility for the study to the Results section. 

Thank you again for the opportunity to resubmit our paper, and for your consideration of our work.

Sincerely,

Sarah Suiter

Sarah.V.Suiter@Vanderbilt.edu

---

## [Decision Letter · Decision Letter 2]

9 May 2024

PONE-D-23-26718R2Utilizing mentorship education to promote a culturally responsive research training environment in the biomedical sciencesPLOS ONE

Dear Dr. Suiter,

Thank you for submitting your manuscript to PLOS ONE. After careful consideration, we feel that it has merit but does not fully meet PLOS ONE’s publication criteria as it currently stands. Therefore, we invite you to submit a revised version of the manuscript that addresses the points raised during the review process.

We look forward to receiving your revised manuscript.

Kind regards,

Delfina Fernandes Hlashwayo, Ph.D.

Academic Editor

PLOS ONE

Journal Requirements:

Reviewers' comments:

Reviewer's Responses to Questions

**Comments to the Author**

1. If the authors have adequately addressed your comments raised in a previous round of review and you feel that this manuscript is now acceptable for publication, you may indicate that here to bypass the “Comments to the Author” section, enter your conflict of interest statement in the “Confidential to Editor” section, and submit your "Accept" recommendation.

Reviewer #2: All comments have been addressed

Reviewer #4: (No Response)

Reviewer #5: (No Response)

2. Is the manuscript technically sound, and do the data support the conclusions?

Reviewer #2: Yes

Reviewer #4: Partly

Reviewer #5: Yes

3. Has the statistical analysis been performed appropriately and rigorously? 

Reviewer #2: Yes

Reviewer #4: N/A

Reviewer #5: Yes

4. Have the authors made all data underlying the findings in their manuscript fully available?

Reviewer #2: Yes

Reviewer #4: Yes

Reviewer #5: Yes

5. Is the manuscript presented in an intelligible fashion and written in standard English?

Reviewer #2: Yes

Reviewer #4: Yes

Reviewer #5: Yes

6. Review Comments to the Author

Reviewer #2: (No Response)

Reviewer #4: This is a well written research article focused on an important and timely topic. The authors present a case study of one institution's implementation of mentor training in culturally competent research mentorship. The authors' have situated their case study within the existing literature on this topic and present a novel perspective from research mentees. There are a couple of issues to note, see below.

Methods:

Focus Groups (Line 177): Who conducted focus groups? Did the interviewees know the interviewer? If so, were there steps taken to alleviate any type of power differential between the interviewer and students?

Data Analysis

(Line 212): The authors mention using an iterative process to code open-ended survey responses, but only explain their first round coding technique.

(line 213): what is the rationale for using GT coding techniques? GT approaches result in generation of a theory and typically are used when the research question is focused on a process, for example "how is mentoring performed at an institution that has institutes culturalkt aware mentor training?" The authors conducted a case study, did not ask questions about process, and do not present a theory or conceptual model. Did S.S. discuss their analytic process/findings with anyone during the analysis phase?

How was qualitative rigor ensured?

Were interviews recorded? Transcribed? How was coding/analysis conducted; did the authors use any qualitative analysis software?

Results-Qualitative

How long did focus groups last?

Conclusions/Limitations

The authors attribute faculty discussing race with students to the mentor training provided at their institution, however qualitative data were collected at a time in U.S. history when race, structural racism, discrimination, and cultural competence were at the forefront of public discussion. This was mentioned in the interview data presented (student discussion of George Floyd). The authors should include the public awareness of these topics during COVID-19 and post-Floyd as part of contextualizing their qualitative data. This public awareness also weakens the interpretation that faculty willingness to discuss race is a result of the training provided.

Reviewer #5: This is my first time reviewing this manuscript. This article studies the self-reported capabilities and opinions of faculty after following a mentor education workshop, including cultural diversity awareness (CDA). The faculty who participated in the workshop was subsequently invited to participate in the study by filling in a survey. Additionally, PhD students participating in the ISMD program whose mentor had participated in the mentor education workshop were invited to participate in three focus group discussions to provide their insights and experiences with mentorship at their institution.

The submitted article is well-written and brings valuable insights to the field of mentorship of PhD students. The mixed-methods approach is sound and valuable, as it gives insights into the self-reported capabilities of mentors and additionally provides insights and experiences of mentees from URM’s. It is imperative that if an organization has requirements of faculty to act as mentors, that mentor education is also provided. The insights and experiences of the mentees can indicate whether the mentor education is adequate and in line with the organization’s goals and policies. These experiences pinpoint specific areas which may need more attention within the organization’s mentoring program, such as ways in which to broach subjects of race/ethnicity and lived experience in relation to work, offering PhD students room to follow their own goals (even when they differ from their mentor’s goals), and offering PhD students room away from work to deal with personal problems.

Although the article is well-written, I do have a few suggestions for minor adjustments which may make the article more easily understandable for the reader.

Abstract: consider including a sentence explaining that faculty participated in a mentorship education intervention, it’s not very explicit as it is.

Methods: line 119, it would help to know the total number of doctoral degrees awarded since 1997 to put the 250 degrees awarded to PEER candidates into context.

Results:

Line 239: ‘Fifty-eight (n=63) percent’ > change to ‘Fifty-eight percent (n=63)’

Line 286: Culturally Diversity Awareness (remove ‘ly’ from Culturally)

Line 297: The authors write that there were two items which significantly differed across the cohorts, but how did these items differ? Were they higher or lower? What does this tell the authors, what conclusions do they draw from this? Please elaborate.

- When looking at the qualitative results from the focus group discussions, there are no real connections made to the workshop followed by faculty. No questions are directly asked about this during the focus group discussions (as mentioned in line 619 of the discussion) because PhD students may not have been aware of their mentor having followed this training. This makes the last line in the introduction (111+112, “…but understanding if these behaviors are implemented and how they are perceived by mentees is an important indicator of institutional change.”) a bit misleading. This line suggests that a connection will be made between the training and the experiences from the mentees. I would suggest removing or rephrasing this last line in the introduction, since we are not able to see how the mentor possibly implemented any lessons learned from the training.

- Lastly as a more general suggestion, I think the article would benefit from a very brief explanation of the role of a mentor for a PhD student at a US university. As a reviewer who has never lived or worked in the US, I am continuously filling in blanks while reading this manuscript about whether or not a PhD student is assigned an official mentor at a US university, whether mentoring is a formal or informal role or task, whether a PhD student’s supervisor is automatically their mentor, whether a mentor needs to be someone in the same research group or department at which the PhD degree is being obtained, etc. By adding a bit more explanation about this, the article would become more easily understandable to a wider audience than only US-based readers. However, this suggestion is optional as I understand it would mean adding length to the paper.

Overall I think the research study is sound, the article is well-written and clear, but would benefit from some minor adjustments, particularly the clarification of the connection between the quantitative and qualitative data.

7. PLOS authors have the option to publish the peer review history of their article (what does this mean?). If published, this will include your full peer review and any attached files.

Reviewer #2: No

Reviewer #4: No

Reviewer #5: **Yes: **Gisela J. van der Velden

---

## [Author Response · Author response to Decision Letter 2]

16 Jun 2024

Sarah V. Suiter, PhD, MS

Human & Organizational Development

Vanderbilt University

230 Appleton Place Peabody #90

Nashville, TN 37203

June 16, 2024

Dear Editors,

My co-authors and I are resubmitting our revised paper, “Utilizing Mentorship Education to Promote a Culturally Responsive Research Training Environment in the Biomedical Sciences” for your review and continued consideration for publication in PLOS One. Thank you for the opportunity to respond to the reviewers’ helpful feedback. Below, we list each reviewer’s suggestions followed by our response to those suggestions and any corresponding changes we have made. In our document uploads, we have included a “clean” version of the edited manuscript, as well as a document with track changes. 

Reviewer #4:

(Line 177): Who conducted focus groups? Did the interviewees know the

interviewer? If so, were there steps taken to alleviate any type of power differential between the

interviewer and students?

● We have added text that address these questions in the methods section 

(Line 212): The authors mention using an iterative process to code open-ended survey

responses, but only explain their first round coding technique.

● We have added text that clarifies our coding strategy for the qualitative survey responses 

(Line 213): what is the rationale for using GT coding techniques? GT approaches result in

generation of a theory and typically are used when the research question is focused on a

process, for example "how is mentoring performed at an institution that has instituted culturally

aware mentor training?" The authors conducted a case study, did not ask questions about

process, and do not present a theory or conceptual model. Did S.S. discuss their analytic

process/findings with anyone during the analysis phase? How was qualitative rigor ensured?

● Grounded theory techniques are well suited to situations in which there is limited understanding of a phenomena; in this case, PEER graduate students’ experiences of mentoring in a predominantly white institution attempting to change institutional culture through mentoring education for faculty. Although we did not present a theory as a result of the analyses we conducted, the grounded theory approach provided a method that allowed us to draw our findings from the codes and themes that emerged from the data, which honored the voices and perspectives of our participants. This emergent process is outlined in our data analysis section. We have added text regarding SS’s analytic process and the use of the research team as critical friends to clarify the techniques we used to enhance rigor. 

Were interviews recorded? Transcribed? How was coding/analysis conducted; did the authors

use any qualitative analysis software?

● We have added text that addresses these questions in the qualitative data analysis section

How long did focus groups last?

● We have added text that addresses the length of the focus groups 

The authors attribute faculty discussing race with students to the mentor training provided at

their institution, however qualitative data were collected at a time in U.S. history when race,

structural racism, discrimination, and cultural competence were at the forefront of public

discussion. This was mentioned in the interview data presented (student discussion of George

Floyd). The authors should include the public awareness of these topics during COVID-19 and

post-Floyd as part of contextualizing their qualitative data. This public awareness also weakens

the interpretation that faculty willingness to discuss race is a result of the training provided.

● Thank you for this suggestion. We have added text to the discussion section contextualizing our data in light of these factors.

Reviewer #5: 

Abstract: consider including a sentence explaining that faculty participated in a mentorship

education intervention, it’s not very explicit as it is.

● We have added a sentence to the abstract explaining that faculty participated in a mentorship education intervention. 

Methods: line 119, it would help to know the total number of doctoral degrees awarded since

1997 to put the 250 degrees awarded to PEER candidates into context.

● We have clarified this paragraph and added the total number of doctoral degrees (among other data) to put PEER degrees in context.

Line 239: ‘Fifty-eight (n=63) percent’ > change to ‘Fifty-eight percent (n=63)’

● We have made this change

Line 286: Culturally Diversity Awareness (remove ‘ly’ from Culturally)

● We have made this change

Line 297: The authors write that there were two items which significantly differed across the

cohorts, but how did these items differ? Were they higher or lower? What does this tell the

authors, what conclusions do they draw from this? Please elaborate.

● We have elaborated on these differences 

When looking at the qualitative results from the focus group discussions, there are no real

connections made to the workshop followed by faculty. No questions are directly asked about

this during the focus group discussions (as mentioned in line 619 of the discussion) because

PhD students may not have been aware of their mentor having followed this training. This makes

the last line in the introduction (111+112, “…but understanding if these behaviors are

implemented and how they are perceived by mentees is an important indicator of institutional

change.”) a bit misleading. This line suggests that a connection will be made between the

training and the experiences from the mentees. I would suggest removing or rephrasing this last

line in the introduction, since we are not able to see how the mentor possibly implemented any

lessons learned from the training.

● Thank you for this comment as we do not intend to mislead our readers. We have revised this sentence to be more clear about what we learned from the focus groups. 

Lastly as a more general suggestion, I think the article would benefit from a very brief

explanation of the role of a mentor for a PhD student at a US university. 

● We have added a brief description of the mentoring relationship 

Thank you again for the opportunity to resubmit our paper, and for your consideration of our work.

Sincerely,

Sarah Suiter

Sarah.V.Suiter@Vanderbilt.edu

---

## [Editor Report · Decision Letter 3]

19 Jun 2024

Utilizing mentorship education to promote a culturally responsive research training environment in the biomedical sciences

PONE-D-23-26718R3

Dear Dr. Suiter,

We’re pleased to inform you that your manuscript has been judged scientifically suitable for publication and will be formally accepted for publication once it meets all outstanding technical requirements.

Kind regards,

Delfina Fernandes Hlashwayo, Ph.D.

Academic Editor

PLOS ONE
---

## [Editor Report · Acceptance letter]

2 Jul 2024

PONE-D-23-26718R3 

PLOS ONE

Dear Dr. Suiter, 

I'm pleased to inform you that your manuscript has been deemed suitable for publication in PLOS ONE. Congratulations! Your manuscript is now being handed over to our production team.

Kind regards, 

on behalf of

Dr. Delfina Fernandes Hlashwayo 

Academic Editor

PLOS ONE